# Association between Drinking Water Nitrate and Adverse Reproductive Outcomes: A Systematic PRISMA Review

**Hannah S. Clausen** [1,*], **Ninna H. Ebdrup** [1,2], **Ida M. Barsøe** [3], **Julie Lyngsø** [4], **Jörg Schullehner** [5,6], **Cecilia H. Ramlau-Hansen** [4], **Bjørn Bay** [2] and **Ulla B. Knudsen** [1,2]

1   Department of Clinical Medicine, Aarhus University, 8200 Aarhus N, Denmark; ninebd@clin.au.dk (N.H.E.); ullaknud@rm.dk (U.B.K.)
2   Department of Obstetrics and Gynaecology, Fertility Clinic, Horsens Regional Hospital, 8700 Horsens, Denmark; bjornbay@me.com
3   Department of Accidents and Emergency, Randers Regional Hospital, 8930 Randers, Denmark; ida.barsoe@gmail.com
4   Department of Public Health, Research Unit for Epidemiology, Aarhus University, 8000 Aarhus C, Denmark; jlyn@ph.au.dk (J.L.); chrh@ph.au.dk (C.H.R.-H.)
5   Department of Public Health, Research Unit for Environment, Work and Health, Aarhus University, 8000 Aarhus C, Denmark; jorg.schullehner@ph.au.dk
6   Geological Survey of Denmark and Greenland (GEUS), 8000 Aarhus C, Denmark
*   Correspondence: hannahsvinth@gmail.com

**Abstract:** One in six couples experience fertility problems. Environmental factors may affect reproductive health; however, evidence is lacking regarding drinking water nitrates and outcomes of male and female fertility. The aim of this study was to investigate if exposure to nitrates in drinking water is associated with adverse reproductive outcomes in humans, and animals of fertile age. We conducted a systematic literature search and included case-control studies, cohort studies, and randomized control trials reporting on the association between drinking water nitrate exposure of men, women, or animals and adverse reproductive outcomes, specified as: Semen quality parameters, time to pregnancy (TTP), pregnancy rates, assisted reproductive technologies (ART), and spontaneous abortion. Findings were reported in a narrative synthesis. A total of 12 studies were included. The only human study included reported a decrease in spontaneous abortion at any detectable nitrate level. Overall, the 11 included animal studies support a potential negative effect on semen quality parameters but report equivocal results on TTP and number of offspring produced, and higher risk of spontaneous abortion. In conclusion, animal studies indicate possible effects on semen quality parameters and spontaneous abortion. However, with a few studies, including some with methodological limitations and small sample sizes, caution must be applied when interpreting these results.

**Keywords:** drinking water; nitrate; adverse reproductive outcomes; subfecundity; fertility; spontaneous abortion; semen quality

## 1. Introduction

The toxicology impact on human health for short and long-term exposure of nitrate is complex and not fully explored. Nitrates occur in both drinking water, food and medicine [1]. The nitrate ion $NO^{3-}$ can undergo transformation to the more potent nitrite ($NO^{2-}$) [2] and to the *N*-nitroso compound (NOC) that are known to be carcinogenic in animals [3] and possibly in humans too [4]. Furthermore,

nitrates can pass the placental blood barrier, thus exposing the fetus in utero [5]. The best evidence for acute and chronic adverse effects of nitrate ingestion via drinking water is its connection with methemoglobinemia, colorectal cancer, thyroid disease, and neural tube defects [4,6,7]. Nitrate drinking water contamination is a global issue, especially in areas with agricultural pressure, and 2–3% of the population in U.S. and Europe might be exposed to levels exceeding the maximum contaminant level (MCL) for drinking water nitrites and nitrates (3 mg/L and 50 mg/L respectively) set by the World Health Organization (WHO) [1,8]. A study of 11 EU countries estimated that 6.5% of the population were exposed to nitrate levels above 25 mg/L nitrate (ranging from 2.0% in the UK to 16.2% in Denmark) [9]. Less data exists from other parts of the world; nonetheless high levels of drinking water nitrate have been reported in India and The Gaza Strip [7]. In areas where nitrate levels in drinking water exceed 50 mg/L, it is estimated to be the most important contributor to nitrate ingestion, but usually it accounts for less than 14% [1]. Nitrate is an important water pollutant, originating from agricultural and other human sources. Depending on the source of nitrate and type of drinking water (ground water or surface water), nitrate concentrations in drinking water vary with season and type of agriculture conducted in the area [1].

The above-mentioned potential genotoxic, teratogenic and carcinogenic properties of nitrate underline the necessity of considering whether environmental aspects, like drinking water nitrates, could explain some of the unknown etiologies of infertility. Approximately 30% of all pregnancies end in spontaneous abortions [10], and approximately one in six couples worldwide experience fertility problems in their reproductive age [11,12]. Part of this is due to low semen quality, to which the etiology is poorly understood. These factors may lead to a prolonged waiting time to pregnancy (TTP) and an increasing need for medically assisted reproduction, which is a burden for the individual and society.

However, only a strikingly low number of studies on possible adverse effects of nitrate and fertility exists. Animal studies have revealed possible adverse effects; in one study, increased spontaneous abortion rate was present in cattle feeding on pastures with high level nitrates [13]. Another showed spontaneous abortion in cattle when ingesting nitrate containing capsules [6]. This suggests that drinking water nitrates might be related to an increased risk of subfecundity and infertility. Furthermore, studies have suggested associations between drinking water nitrates and other adverse reproductive outcomes, e.g., stillbirth, preterm birth, low birth weight, small for gestational age (SGA) and most convincing birth defects of the central nervous system [7].

The association between drinking water nitrates and human health has previously been reviewed [3,6,7,14,15]. A systematic focus on reproductive health outcomes including human male and female studies as well as animal studies is, however, absent.

Therefore, the aim of this study was to investigate whether exposure to nitrates in drinking water is associated with adverse reproductive outcomes in men, women, or animals of fertile age.

## 2. Methods

We performed at systematic search of literature describing associations between exposure to nitrates in drinking water with adverse reproductive outcomes in men, women, and animals of fertile age. The outcomes assessed were measures of subfecundity and infertility (e.g., semen quality parameters, TTP, pregnancy rates, use of assisted reproductive technologies (ART)), and spontaneous abortion, which could be an indirect measurement of infertility. The included studies were case-control studies, cohort studies, and randomized control trials (RCT).

This systematic review follows the Preferred Reporting Items for Systematic Reviews and Meta-Analyses (PRISMA) statement [16], a protocol and a flowchart was made accordingly hereto. Elaborated reasons for exclusion are available in Supplementary Material.

### 2.1. Search Strategy and Study Selection

A systematic computerized literature search was conducted on 2 May 2019 using the databases PubMed/Medline, Embase, and The Cochrane Library. The search strategy was developed and conducted by two authors (H.S.C., N.H.E.) in cooperation with a medical librarian.

To identify potentially relevant studies' keywords, i.e., medical subject headings (MeSH), and Emtree terms were used. Additionally, a search using free text terms was conducted to include new, non-indexed literature. The search was made with a restriction to the English language but no limit on year of publication. Furthermore, the bibliographies of the included studies were hand searched and other studies citing the included studies were searched for in Scopus in order to include additional relevant studies.

From the searches, all retrieved studies were screened by title and abstract for eligibility individually by two authors (H.S.C., N.H.E.), reaching consensus by discussion including a third reviewer (U.B.K.) if necessary. Studies potentially eligible for inclusion were retrieved and read in full text by two authors (H.S.C., N.H.E.) to ensure they met the following inclusion criteria:

1.  Studies containing a relevant population:

    a.  Women of fertile age (15–51 years)
    b.  Men of fertile age (15–60 years)
    c.  Animal population

2.  Studies reporting a numerical exposure range for drinking water nitrate
3.  Studies containing a relevant control group
4.  Studies investigating at least one of the outcomes:

    a.  Subfecundity or fertility (TTP, pregnancy rates, assisted reproductive technology treatment (ART))
    b.  Spontaneous abortion
    c.  Semen quality

5.  Original studies
6.  Studies with one of the following designs:

    a.  Case-control study
    b.  Cohort study
    c.  Randomized control trials (RCT) study

Studies not meeting the criteria mentioned above were excluded. Agreement was reached through discussion by two authors (H.S.C., N.H.E.) and, if necessary, a third author was consulted.

The computerized literature search was repeated on 25 November 2019 and no new literature was deemed eligible for inclusion.

### 2.2. Data Extraction and Quality Assessment

All included studies were read in full text. To ensure a standardized procedure, data was extracted using an a priori specified data extraction form, available in Supplementary Material. The quality was assessed by the Newcastle-Ottawa Scale (NOS) [17] for human studies and the Risk of Bias (RoB) tool by Systematic Review Centre for Laboratory animal Experimentation (SYRCLE) [18] for animal studies. The assessment was performed separately by two authors (H.S.C., N.H.E.), reaching agreement by discussion and involving a third author (U.B.K.) if necessary.

SYRCLE's RoB tool is based on the Cochrane Collaboration RoB tool [19] and focuses on aspects of risk of bias that are relevant when considering animal studies. All the included animal studies were assessed a score of high, unclear or low risk of bias according to the ten items of SYRCLE's RoB tool.

　　　To ensure standardized scoring, an explanatory form was made for NOS. "Maternal age" was chosen to be the most important covariate in human studies. Studies were allocated a score between 0 and 9. The studies with a score of 7 or above were considered high quality studies and they were observed separately to see if they had any impact on the conclusion of this systematic review.

　　　No core outcome set (COS) was available for the outcomes in this review. No meta-analyses were made. Regarding Aschengrau et al. 1989 [20], the concentration reported in the article was corrected from mg/L to mg-N/L after personal communication. Apart from the above mentioned, no authors, investigators and alike were contacted to obtain missing information, nor were protocols for included studies obtained.

## 3. Results

　　　A total number of 144 potentially relevant studies were found through database and bibliography searches. Of these, 12 were found eligible for inclusion in the review (Figure 1).

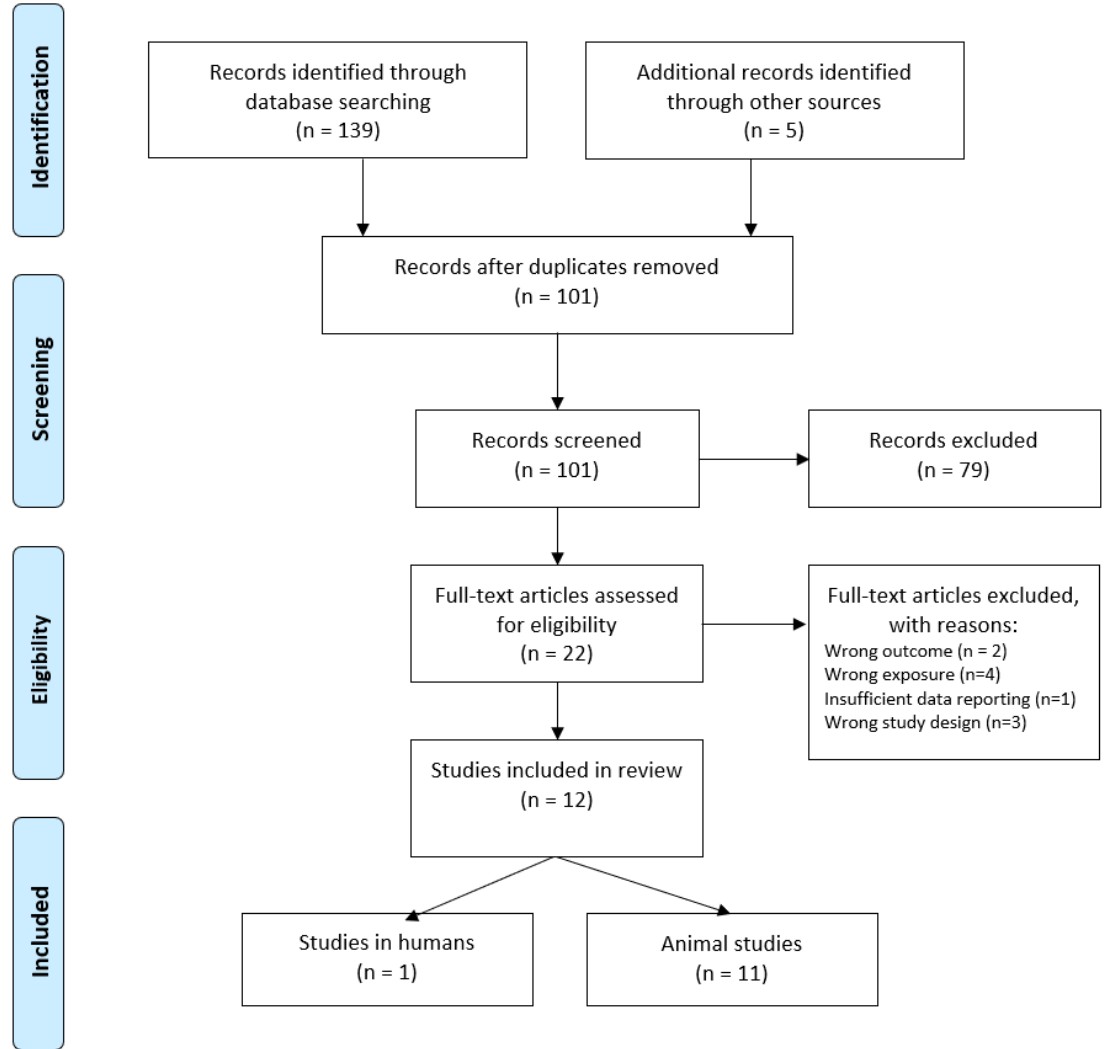

**Figure 1.** Flowchart of literature search and identification.

　　　Study characteristics, results, and the assigned score from quality assessment are presented for the human studies in women (Table 1), animal studies in females (Table 2) and animal studies in males (Table 3). The reported nitrate values and forms represent the original from the included studies; hence, no conversion was made except for Aschengrau et al. 1989 [20].

**Table 1.** Studies investigating adverse reproductive outcomes in women included in the systematic review.

| First Author, Year, Country | Study Design | Study Population (Size and Selection) | Exposure Description | Outcome | Control for Maternal Age | Control for Other Confounding Factors | Effect of Nitrate Exposure | Main Results | Total NOS Score |
|---|---|---|---|---|---|---|---|---|---|
| Aschengrau, 1989, US [20] | Case-control | 286 cases, 1391 controls | Public water supply<br><br>1: Undetectable level of nitrate (<0.1 mg-N/L) or nitrite (<0.01 mg-N/L)<br><br>2: Any detectable level of nitrate (0.1–5.5 mg-N/L)<br><br>3: Any detectable level of nitrite (0.01–0.03 mg-N/L) | Spontaneous abortion (<28 weeks gestation [a]) | Yes | Water quality, educational level, history of prior spontaneous abortion. | ↓ | Results from any detectable level: All subjects, crude: Nitrate: OR (95% CI) = 0.4 (0.3, 0.6); Nitrite: OR (95% CI) = 1.1 (0.8, 1.6)<br><br>All subjects, adjusted: Nitrate: OR (95%CI) = 0.5 (0.2, 0.9) | 9 |

Abbreviations: NOS = Newcastle Ottawa Scale. OR = Odds Ratio. CI = Confidence Interval. ↓: decreased risk of adverse reproductive outcomes. [a]: Cut-off between spontaneous abortion and stillbirth as stated in the study but this definition has changed over time.

**Table 2.** Animal studies investigating adverse reproductive outcomes in females included in the systematic review.

| First Author, Year, Country | Study Design | Animal Age, Species and Source | Groups | Study Duration | Outcome | Control for Feed and/or Water Consumption | Control for Other Factors | Effects of Nitrate Exposure | Main Results |
|---|---|---|---|---|---|---|---|---|---|
| Anderson, 1978, US [21] | RCT | 5–6 weeks old female and male swiss CD-1 mice from breeding farm | Drinking water containing: Experiment 1: 1: Control group (10 females); 2: NaNO$_2$ (1 g/L/1000 ppm) in drinking water (10 females) Experiment 2: 1: Control group (20 females) 2: NaNO$_2$ (1 g/L/1000 ppm) in drinking water (20 females) | Months | Experiment 1: No. of females with surviving litters, no. of offspring.<br><br>Experiment 2: Conception time, infertility, litter size, rates of stillbirth, neonatal death. | Yes | Housing (environmental and hygienic conditions, no. per cage) | ↑ | Experiment 1: Lower no. offspring in treated group; 30% vs. 0% in control group had no surviving litters ($p < 0.01$). Experiment 2: On average 5 days longer to conception, 3 more mice littered more than 30 days after males were added, smaller litters. None of these results had $p < 0.05$ comparing to control group. |
| Anderson, 1985, US [22] | RCT | 7–8 weeks old female C57BL/6 and male BALB/c mice from laboratory | Drinking water containing:<br><br>1: Control, no treatment (66 females, 51 males)<br><br>2: 184 ppm NaNO$_2$ in drinking water (39 females, 52 males)<br><br>3: 1840 ppm NaNO$_2$ in drinking water (65 females, 54 males) | Median 27 months (depending on survival) | No. of litters, effective no., number of females becoming pregnant, average time from introduction of the male until birth, no. of stillborn litters, average litter size at birth. | Yes | Housing (bedding, environmental conditions) | (↑) | No. of litters: 20 in control group, 14 in 184 ppm group, 15 in 1840 ppm group. Data not shown for the other outcomes. All outcomes described as non-significant. |

**Table 2.** *Cont.*

| First Author, Year, Country | Study Design | Animal Age, Species and Source | Groups | Study Duration | Outcome | Control for Feed and/or Water Consumption | Control for Other Factors | Effects of Nitrate Exposure | Main Results |
|---|---|---|---|---|---|---|---|---|---|
| Greenlee, 2004, US [23] [a] | RCT | Mice embryos from CD-1 female mice (21–26 days old), from laboratory. Incubated in vitro | Drinking water containing: 1: Control, 0.1% ethanol (20–25 embryos) 2: 1 ppm $NH_4NO_3$ in drinking water, which is based on RfD [b] (20–25 embryos) | 96 h (corresponding to the first 5–7 days after human conception) | Development to blastocyst, percentage of apoptosis, mean cell number per embryo. | Not applicable | Same incubation conditions | ↑ | $NH_4NO_3$: Reduced mean cell number per embryo ($p \leq 0.0005$), increased percentage of apoptosis ($p \leq 0.05$) compared to control. |
| National Toxicology program, 2001, US [24] | RCT | Male and female B6C3F1 mice from animal farm, average age 6 weeks | Drinking water containing: 1: 0 ppm $NaNO_3$ (10 males, 10 females) 2: 375 ppm $NaNO_3$ (10 males, 10 females) 3: 750 ppm $NaNO_3$ (10 males, 10 females) 4: 1500 ppm $NaNO_3$ (10 males, 10 females) 5: 3000 ppm $NaNO_3$ (10 males, 10 females) 6: 5000 ppm $NaNO_3$ (10 males, 10 females) | 98 days | Estrous cycle length, estrous stages. | Yes | Weight at baseline, Housing (no. per cage, environmental and hygienic conditions) | ↑ | Estrous cycle length was prolonged in 1500 ppm ($p \leq 0.05$) and 5000 ppm ($p \leq 0.01$) groups. |
| National Toxicology program, 2001, US [24] | RCT | Male and female F344/N rats from animal farm, average age 7 weeks | Drinking water containing: 1: 0 ppm $NaNO_3$ (10 males, 10 females) 2: 375 ppm $NaNO_3$ (10 males, 10 females) 3: 750 ppm $NaNO_3$ (10 males, 10 females) 4: 1500 ppm $NaNO_3$ (10 males, 10 females) 5: 3000 ppm $NaNO_3$ (10 males, 10 females) 6: 5000 ppm $NaNO_3$ (10 males, 10 females) | 98 days | Estrous cycle length, estrous stages. | Yes | Weight at baseline, Housing (no. per cage, environmental and hygienic conditions) | → | Non. |
| Sleight, 1968, US [25] | RCT | Female and male guinea pigs of unknown age | Drinking water containing: 1: 0 ppm $KNO_3/KNO_2$ 2: 300 ppm $KNO_3$ 3: 2500 ppm $KNO_3$ 4: 10,000 ppm $KNO_3$ 5: 30,000 ppm $KNO_3$ 6: 300 ppm $KNO_2$ 7: 1000 ppm $KNO_2$ 8: 2000 ppm $KNO_2$ 9: 3000 ppm $KNO_2$ 10: 4000 ppm $KNO_2$ 11: 5000 ppm $KNO_2$ 12: 10,000 ppm $KNO_2$ 3–6 females per group + at least 1 male | 100–240 days | No. of litters, relative percent reproductive performance, aborted fetuses, mummified or absorbed fetuses, stillborn fetuses, percent of fetal loss. Tissue samples from: Ovaries, uterus, cervix, thyroid. | Yes | Housing (no. per cage) | (↑) | $KNO_3$: Poor reproductive performance at 30,000 ppm. $KNO_2$: No live births were seen at 5000 or 10,000 ppm levels due to abortion or mummification. Dose-response relationship. |

Abbreviations: RCT = Randomized Controlled Trial; $NaNO_2$ = Sodium nitrite; $NH_4NO_3$ = Ammonium nitrate; $NaNO_3$ = Sodium nitrate; $KNO_3$ = Potassium nitrate; $KNO_2$ = Potassium nitrite. ↑: significant ($p < 0.05$) increased risk of adverse reproductive outcomes. (↑): non-significant ($p > 0.05$) increased risk of adverse reproductive outcomes. →: no difference between exposed and non-exposed groups. a: study was conducted on embryos which are probably both male and female. The study is reported in the table for female outcomes, because the outcome measured is similar to effect on fetus (e.g., abortion) which is usually measured on females. b: RfD (reference dose) is an estimate of a daily oral exposure to human population (including sensitive subgroups) that is likely to be without an appreciable risk.

Table 3. Animal studies investigating adverse reproductive outcomes in males included in the systematic review.

| First Author, Year, Country | Study Design | Animal Age, Species and Source | Groups | Study Duration | Outcome | Control for Feed and/or Water Consumption | Control for Other Factors | Effects of Nitrate Exposure | Main Results |
|---|---|---|---|---|---|---|---|---|---|
| Aly, 2010, Egypt [26] | RCT | Adult male swiss albino rats from animal facility | Drinking water containing: 1: 0 mg mg/kg/day $NaNO_3$ (6 males) 2: 50 mg/kg/day $NaNO_3$ (6 males) 3: 100 mg/kg/day $NaNO_3$ (6 males) 4: 200 mg/kg/day $NaNO_3$ (6 males) | 60 days | Sperm: Sperm count, sperm motility, daily sperm production. Testis: testicular lactic dehydrogenase-X (LDH-X), glucose-6-phosphatate dehydrogenase (G6PD), acid phosphatase (AP), testis weight, histopathological examination of testis, hydrogen peroxide generation ($H_2O_2$), lipid peroxidation (LPO), antioxidant enzymes activity, reduced glutathione (GSH). | No | Weight at baseline, housing ("standard laboratory conditions") | ↑ | Significant ($p < 0.05$) negative effect in a dose-response relationship was seen in all treatment groups compared to controls on: sperm count, sperm motility, daily sperm production, testicular enzymes activity, hydrogen peroxide generation ($H_2O_2$), lipid peroxidation (LPO), reduced glutathione (GSH). Significant ($p < 0.05$) lower testis weight in treatment group: 100 and 200 mg/kg/day compared to controls. Induction of histopathological changes. Antioxidant activity significantly ($p < 0.05$) decreased in some groups compared to controls. |
| Amini, 2016, Iran [27] | RCT | Adult male mice from animal care unit | Drinking water containing: 1: 0 mg/L $NaNO_2$ (5 males) 2: 3 mg/L $NaNO_2$ (5 males) 3: 10 mg/L $NaNO_2$ (5 males) 4: 50 mg/L $NaNO_2$ (5 males) | 60 days | Expression of laminin $\alpha5$ in basal level, middle level and elongated spermatid (apical compartment) of seminiferous epithelium. | No | Weight at baseline, housing ("controlled environment") | ↑ | Basal and middle levels of seminiferous epithelium: Differences in expression of laminin but not significant. Elongated spermatid of seminiferous epithelium: Dose-response relationship difference in expression, significant ($p = 0.032$) increase in 50 mg/L treatment group compared to control. Real time PCR showed significant ($p = 0.001$) increase in laminin ratio in control group compared to experimental groups. |

**Table 3.** *Cont.*

| First Author, Year, Country | Study Design | Animal Age, Species and Source | Groups | Study Duration | Outcome | Control for Feed and/or Water Consumption | Control for Other Factors | Effects of Nitrate Exposure | Main Results |
|---|---|---|---|---|---|---|---|---|---|
| Amini, 2017, Iran [28] | RCT | Fertile male mice | Drinking water containing:<br>1: Control, free of contaminants (6 males)<br>2: 3 mg/L NaNO$_2$ (6 males)<br>3: 50 mg/L NaNO$_2$ (6 males) | 60 days | Expression of laminin α5 as total amount in testicular tissue and in extracellular matrix of mice testicular interestitium. | No | No information | ↑ | Significant ($p = 0.003$) reduction in total level of laminin in testicular tissue in 50 mg/L group. No changes in expression of laminin in extracellular matrix. |
| Amini, 2018, Iran [29] | RCT | Fertile male mice from animal house | Drinking water containing:<br>1: Control, pollutant-free (8 males)<br>2: 3 mg/L NaNO$_2$ (8 males)<br>3: 50 mg/L NaNO$_2$ (8 males) | 60 days | Testis weight. Expression of fibronectin in testicular interstitial tissue. | No | Weight at baseline, housing ("suitable conditions") | (↑) | Insignificantly ($p = 0.094$) weaker expression of fibronectin in 50 mg/L group. |
| Attia, 2013, Egypt [30] | RCT | 16 weeks old New Zealand white male rabbits | Drinking water containing:<br>1: 14 ppm/tap water (7/4/3 males)<br>2: 350 ppm NaNO$_3$ (7/4/3 males)<br>3: 700 ppm NaNO$_3$ (7/4/3 males) | 322 days | Sperm: Sperm concentration, total sperm, total live sperm, total dead sperm, total abnormal sperm. Age at first ejaculate, male fertility, number of offspring, testosterone (seminal and blood plasma), histopathology of testis. | Yes | Weight at baseline, housing (no. per cage, environmental and hygienic conditions) | ↑ | Significant negative effect was seen in 700 ppm group compared to 350 ppm and control (dose-response relationship) on: Blood plasma testosterone ($p = 0.004$), seminal plasma testosterone ($p = 0.0001$), sperm concentration ($p = 0.03$), total sperm output ($p = 0.008$), total live sperm ($p = 0.002$), total dead sperm ($p = 0.006$), total abnormal sperm ($p = 0.01$), age at first ejaculate ($p = 0.01$), fertility ($p = 0.03$), no. of offspring ($p = 0.02$). |
| National Toxicology Program, 2001, US [24] | RCT | Male and female B6C3F1 mice from animal farm, average age 6 weeks | Drinking water containing:<br>1: 0 ppm NaNO$_3$ in (10 males, 10 females)<br>2: 375 ppm NaNO$_3$ (10 males, 10 females)<br>3: 750 ppm NaNO$_3$ (10 males, 10 females)<br>4: 1500 ppm NaNO$_3$ (10 males, 10 females)<br>5: 3000 ppm NaNO$_3$ (10 males, 10 females)<br>6 5000 ppm NaNO$_3$ (10 males, 10 females) | 98 days | Organ weights: Cauda epididymis, epididymis, testis. Sperm: Spermatid heads, spermatid count, sperm motility, sperm concentration. | Yes | Weight at baseline, housing (no. per cage, environmental and hygienic conditions) | ↑ | Sperm motility in 5000 ppm ($p \leq 0.01$) males was significantly decreased. |

**Table 3.** *Cont.*

| First Author, Year, Country | Study Design | Animal Age, Species and Source | Groups | Study Duration | Outcome | Control for Feed and/or Water Consumption | Control for Other Factors | Effects of Nitrate Exposure | Main Results |
|---|---|---|---|---|---|---|---|---|---|
| National Toxicology Program, 2001, US [24] | RCT | Male and female F344/N rats from animal farm, average age 7 weeks | Drinking water containing: 1: 0 ppm $NaNO_3$ in (10 males, 10 females) 2: 375 ppm $NaNO_3$ (10 males, 10 females) 3: 750 ppm $NaNO_3$ (10 males, 10 females) 4: 1500 ppm $NaNO_3$ (10 males, 10 females) 5: 3000 ppm $NaNO_3$ (10 males, 10 females) 6 5000 ppm $NaNO_3$ (10 males, 10 females) | 98 days | Organ weights: Cauda epididymis, epididymis, testis. Sperm: Spermatid heads, spermatid count, sperm motility, sperm concentration. | Yes | Weight at baseline, housing (no. per cage, environmental and hygienic conditions) | ↑ | Sperm motility lower in 1500 ppm ($p \leq 0.05$) and 5000 ppm ($p \leq 0.01$) groups. Epididymis weight lower in 5000 ppm group ($p \leq 0.05$). |
| Pant, 2002, India [31] | RCT | 7 weeks old male Swiss white mice from animal colony | Drinking water containing: 1: Control, tap water (5 males) 2: 90 ppm $KNO_3$ (5 males) 3: 200 ppm $KNO_3$ (5 males) 4: 500 ppm $KNO_3$ (5 males) 5: 700 ppm $KNO_3$ (5 males) 6: 900 ppm $KNO_3$ (5 males) | 35 days | Organ weights: Testis, epididymis, seminal vesicle, ventral prostate, coagulating gland. Sperm: Sperm count, sperm motility, morphological abnormalities in sperm. Testicular enzymes activity: Sorbitol dehydrogenase (SDH), lactate dehydrogenase (LDH), 17-β hydroxysteroid dehydrogenase (17-βHSD), acid phosphatase (AP), β glucuronidase (β-G), γ glutamyl transpeptidase (γ-GT). | Yes | Weight at baseline, Housing ("standard laboratory conditions") | ↑ | Significant ($p < 0.05$) effect in 900 ppm exposure group; declined activity of 17-β hydroxysteroid dehydrogenase (17-βHSD, increased activity of γ glutamyl transpeptidase (γ-GT), decrease in sperm count and motility, increase in total abnormal sperm. |

**Table 3.** *Cont.*

| First Author, Year, Country | Study Design | Animal Age, Species and Source | Groups | Study Duration | Outcome | Control for Feed and/or Water Consumption | Control for Other Factors | Effects of Nitrate Exposure | Main Results |
|---|---|---|---|---|---|---|---|---|---|
| Sleight, 1968, US [25] | RCT | Female and male guinea pigs of unknown age | Drinking water containing: 1: 0 ppm $KNO_3/KNO_2$ 2: 300 ppm $KNO_3$ 3: 2500 ppm $KNO_3$ 4: 10,000 ppm $KNO_3$ 5: 30,000 ppm $KNO_3$ 6: 300 ppm $KNO_2$ 7: 1000 ppm $KNO_2$ 8: 2000 ppm $KNO_2$ 9: 3000 ppm $KNO_2$ 10: 4000 ppm $KNO_2$ 11: 5000 ppm $KNO_2$ 12: 10,000 ppm $KNO_2$ 3–6 females per group + at least 1 male | 100–240 days | Reproductive performance. | Yes | Housing (no. per cage) | → | Conception at all levels of treatment; male fertility apparently not impaired. |

Abbreviations: RCT = Randomized Controlled Trial; $NaNO_3$ = Sodium nitrate; $NaNO_2$ = Sodium nitrite; $NH_4NO_3$ = Ammonium nitrate; $KNO_3$ = Potassium nitrate; $KNO_2$ = Potassium nitrite. ↑: significant increased risk of adverse reproductive outcomes. (↑): non-significant increased risk of adverse reproductive outcomes. →: no difference between exposed and non-exposed groups.

*3.1. Studies Reporting on Human Outcomes*

One human study reporting on female outcomes [20] was included. This American case-control study compromised 286 case subjects and 1391 control subjects. It reported on spontaneous abortion up to 28 weeks of gestation and found a decreased risk of spontaneous abortion with an adjusted OR (95% CI) of 0.5 (0.2, 0.9) at any detectable nitrate level (0.1–5.5 mg/L) and a crude OR (95% CI) of 1.1 (0.8, 1.6) at any detectable nitrite level (0.01–0.03 mg/L). Comparison was made with undetectable levels and measured in public water supply. No studies investigating TTP, pregnancy rates, impact on ART-results, or male reproductive outcomes in humans were found.

*3.2. Studies Reporting on Animal Outcomes*

3.2.1. Studies on Female Animals

Five of the included studies reported female outcomes [21–25]. Three of these studies [21,22,25] reported on longer days to litter (comparable to extended TTP) and/or fewer offspring produced (comparable to pregnancy rate). Anderson et al. 1978 [21] reported on this in mice exposed to 1000 ppm sodium nitrate in drinking water compared to controls. In the study by Anderson et al. 1985 [22], mice exposed to 184 and 1840 ppm sodium nitrate had a lower number of litters compared to controls not being exposed to sodium nitrate. Sleight and Atallah 1968 [25] investigated on guinea pigs exposed to potassium nitrate or potassium nitrite compared to controls. They found a poor reproductive performance at an exposure of 30,000 ppm potassium nitrate, no live births at 5000 or 10,000 ppm potassium nitrite, and overall a dose-response relationship. Higher rates of spontaneous abortion/fetal death or increased apoptosis in the embryo were reported in three studies [21,23,25]. All the animal studies showed an inverse association between nitrate levels and minimum one outcome in at least one exposure group, primarily the highest exposure groups. In one study [23], an increased apoptosis in mice embryos was seen at low doses not expected to be harmful to humans.

3.2.2. Studies on Male Animals

Eight studies [24–31] reported on male outcomes. The outcomes were reported differently across the studies with direct and indirect measurements of semen parameters. Possible negative effects on semen parameters were suggested in all included studies except for one [25], which reported no association as an indirect measurement, e.g., conception in guinea pigs was seen at all levels of exposure but this was not specified in detail as the female outcomes were the focus of this study.

Two studies studied rats. One [24] reported on lower sperm motility and lower epididymis weight, the other [26] reported significant reductions of e.g., sperm count, sperm motility, testicular enzymes, and testis weight.

Five studies studied mice. Expression of laminin $\alpha$5 (a glycoprotein) in testicles was studied in two of these studies and changes were observed in the seminiferous epithelium [27] and in total level in testicular tissue but not in the extracellular matrix (ECM) [28]. One study in mice [29] found weaker expression of fibronectin in testis. Two studies [24,31] found no differences in mice organ weights (e.g., testis and epididymis) but both of these studies found a reduced sperm motility and one [31] also reduced sperm count, increased total abnormal sperm and declined activity of testicular enzymes. One study [30] on rabbits revealed an inverse association between nitrate exposure and e.g., testosterone levels, sperm parameters and number of offspring.

*3.3. Quality of Included Studies*

The study in humans [20] obtained a NOS score of 9 and was therefore considered a high quality study.

The quality of the animal studies was varied when evaluated with SYRCLE's RoB tool, and were mainly low quality, as the score "unclear risk of bias" was dominant, with all rated studies assigned

this score in five out of ten score-items due to weak reporting. A score of "high risk of bias" was given once to two studies [23,30].

Further details on quality assessment can be found in the Supplementary Material.

## 4. Discussion

This systematic review reports findings in human and animal studies on the potential influence of nitrate in drinking water in relation to adverse reproductive outcomes. It reveals that only few studies have been conducted in this field and highlights the complexity in evaluating the exposure of nitrates. Animal studies support a possible association between exposure to drinking water nitrates and semen quality parameters and spontaneous abortion. Only one study investigating human outcomes was found, reporting a lowered risk of spontaneous abortions in women exposed to drinking water nitrates.

### 4.1. Strengths and Limitations

This review is the first to systematically review studies on the potential effects of exposure to drinking water nitrates and adverse reproductive outcomes with focus on fertility measures. Despite the comprehensive literature search, there might be incomplete retrieval of data due to publication bias and the English language restriction. This problem was partly addressed as additional literature was searched for in the bibliographies of included studies.

The lack of consensus on the definition of spontaneous abortion in the included studies may be another limitation. UpToDate defines spontaneous abortion as fetal loss up to 20 weeks of gestation [32], in Denmark the limit is set at 22 weeks of gestation [33] and the study by Aschengrau et al. 1989 set 28 weeks of gestation as the limit in 1989 [20]. The 28-week limit is consistent with a change of definition over time when considering the older date of the study [20]. The different limits in gestational age demonstrate that there may be an overlap between the outcome spontaneous abortion and perinatal outcomes like stillbirth, which were not evaluated in this review. Similarly, there was no defined cut-off available in the animal studies to separate spontaneous abortion from perinatal outcomes.

### 4.2. Studies Reporting on Human Outcomes

Overall, the only human study was from 1989, thus revealing a lack of focus on this area. The study [20] showed an OR below 1 for spontaneous abortion at any detectable nitrate level, thus pointing to a possible beneficial effect. This aligns with the possible beneficial effect of nitrates on blood pressure [1]. The nitrate levels reported in this study are low compared to the maximum contaminant level (MCL) for drinking water nitrites and nitrates 3 mg/L and 50 mg/L respectively set by the World Health Organization (WHO) [1]. The study was high quality (NOS score of 9), but limitations were still present due to e.g., the high complexity of evaluation of nitrate exposure.

It can be questioned whether the composition in the public water taps reflected the actual composition of the drinking water ingested by the women, as the amount of home water intake, and respectively bottled water intake, was not considered and these factors vary between individuals [34]. Furthermore, water samples close to the pregnancy outcome were not always available, and it can be discussed whether the nitrate levels in groundwater were stable over time [35].

Aschengrau et al. 1989 [20] did not report details on other nitrate sources (e.g., food or medicine), nor did the study account for the endogenous nitrosation of nitrates. A follow-up study from 2019 suggested an association between increased risk of stillbirth and use of nitrosable drugs of which some are very commonly used during pregnancy [36].

Aschengrau et al. 1989 [20] adjusted for some other water contaminants, but still it is possible that unmeasured water contaminants could explain some of the results.

No studies were found to report on the important human outcomes: Subfecundity or infertility (specified as extended TTP), lower pregnancy rates, and use of ART, spontaneous abortion, or semen quality parameters.

Several studies did not qualify for inclusion in this systematic review. These, however, have indications of the possible influence of nitrate on fertility. A review on studies on tap water indicated some correlation with spontaneous abortion [37], but the exposure of nitrates was not specified. A case-report [38] from a U.S. community reported a cluster of spontaneous abortions in women exposed to high drinking water nitrate levels. In contrast, a cross-sectional study [39] showed higher infant mortality but no higher rate of spontaneous abortion or stillbirths in mothers living in high-nitrate areas in West Africa. Similarly, an ecological study [40] from the U.S. including 30,980 infants and fetuses showed no association between drinking water nitrates and fetal mortality.

Extensive evidence exists on the association between drinking water nitrates and thyroid disease, development of infant methemoglobinemia (an affection of oxygen transportation) and fetal malformations. This is also what WHO primarily aims to avoid with the MCL for nitrates [1]. The above mentioned health aspects and underlying mechanisms could also be indirectly related to infertility and fetal death [25,38,41–43]. In line with this, a U.S. study of 25 women [44] showed a possible relationship between high maternal methemoglobinemia level and spontaneous abortion in the first trimester.

### 4.3. Studies Reporting on Animal Outcomes

The animal studies reported different outcomes and scored by SYRCLE's RoB tool, they were of varying quality, indicating low quality. The risk of bias was classified as "unclear" according to several parameters in SYRCLE's RoB tool. This was mainly due to a lack of reporting on or actual blinding of investigators and caretakers or randomization. These factors might be overlooked as important factors due to the similarity in appearance that is often present in the animals studied. In general, there seems to be a lack of consensus and tradition on how to report animal studies and what to include. The Navigation Guide Methodology [45] and The REFLECT statement [46] was made to address this issue and introduce new methods for environmental health research, but still needs further implementation.

The animals showed equivocal results on days to litter and number of offspring produced and negative results on spontaneous abortion and fetal death. Overall, a negative association between drinking water nitrates and semen quality parameters in animal studies was found. Nonetheless, none of the studies were described as blinded, which could lead to an overestimation of the potential effect on the semen quality parameters, because the investigators might be biased. Furthermore, assessing the effect in the animal studies is challenged by the heterogenicity regarding reported outcomes and animal species.

The advantages of the animal studies are the more exact diet and water consumption, the relatively similar animals and controlled environment which make it less complex to evaluate the actual nitrate exposure. This also makes it possible to study other aspects of nitrate exposure as done in some of the included animal studies. Anderson et al. 1978 [21] showed a lower number of offspring and a higher number of infertile female mice in the group exposed to the drug imipramine (a tricyclic antidepressant) together with nitrate compared to nitrate exposure alone. Another included animal study [22] showed similar litter sizes in groups exposed to nitrate alone or in combination with the nitrosable drug cimetidine. Furthermore, Attia et al. 2013 [30] showed some reversing effect of adding antioxidants and probiotics to the nitrate contaminated water, pointing to the importance of taking into considerations the amount and type of food consumed.

Evaluating drinking water nitrates without taking into consideration other drinking water contaminants might be problematic. The included study on mice embryos [23] accounted for this by making an exposure group with ammonium nitrate alone and one with a mixture of groundwater contaminants; both exposures resulting in significantly reduced cell numbers in embryos.

Among studies excluded from this review, one assimilated a realistic mixture of groundwater contamination and results on female fertility in rats and mice were equivocal [47]. Another study, showed indication of a protective effect of nitrate in drinking water on semen parameters in diabetic

mice [48], while two other studies [49,50] showed no association between nitrate in drinking water and litter size or fertility.

*4.4. Comparisons: Studies in Humans and Animals*

Animal studies are indicators of possible mechanisms in humans, but many challenges exist when transferring these results to actual effects in humans. Animal studies will often be conducted with a high exposure over a shorter period (acute toxic dose)—which is also the case for the studies included in this review. As opposed to this, humans are often exposed to a lower exposure for a longer period. Considering the difference in exposure dose and time, the mechanisms might differ. Furthermore, one cannot directly transfer study results from one species of animal to another, nor from animals to humans or from humans living in different exposure settings (i.e., external validity). Given these issues, the MCL set by WHO has been questioned [51]. Thus, in future research these factors should be considered.

**5. Conclusions**

In conclusion, this systematic review showed that equivocal and scarce evidence of the impact of nitrates in drinking water on adverse reproductive outcomes exists. Only one study in humans was found, showing lower rate of spontaneous abortion at higher exposure levels. No studies were found reporting on the important human outcomes subfecundity, infertility, and semen parameters. The included animal studies reported inverse associations between nitrate exposure and semen parameters and equivocal results regarding female fertility. However, studies were scored to be of low quality.

Large high-quality epidemiological studies are needed to investigate the possible effects suggested by the animal studies. These studies should account for the complexity of evaluating nitrate exposure and include the outcome measures spontaneous abortion, extended TTP and semen quality parameters.

**Supplementary Materials:** The following are available online at http://www.mdpi.com/2073-4441/12/8/2287/s1, Table S1: Characteristics and details on quality assessment of studies on human outcomes according to Newcastle Ottawa Scale, Table S2: Results for quality assessment of animal studies using SYRCLE's Risk of Bias tool.

**Author Contributions:** Conceptualization, H.S.C., N.H.E., I.M.B., J.S., C.H.R.-H., and U.B.K.; methodology, H.S.C., N.H.E., I.M.B., J.L., B.B. and U.B.K.; validation, H.S.C. and N.H.E.; formal analysis, H.S.C., N.H.E.; investigation, H.S.C., N.H.E.; data curation, H.S.C., N.H.E.; writing—original draft preparation, H.S.C., N.H.E., I.M.B. and U.B.K.; writing—review and editing, H.S.C., N.H.E., I.M.B., J.L., J.S., C.H.R.-H., B.B., and U.B.K.; visualization, H.S.C.; supervision, U.B.K.; project administration, H.S.C., N.H.E, U.B.K. All authors have read and agreed to the published version of the manuscript.

**Funding:** This research received no external funding.

**Conflicts of Interest:** B.B. reports personal fees from Merck (lecture for fertility staff in offspring health following ART), outside the submitted work. This had no role in the design of the study; in the collection, analyses, or interpretation of data; in the writing of the manuscript, or in the decision to publish the results.

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
