# Peer review of "Association between Drinking Water Nitrate and Adverse Reproductive Outcomes: A Systematic PRISMA Review"

_water, doi:10.3390/w12082287_

Round 1
Reviewer 1 Report
The paper by Clausen et al entitled 'Association between Drinking Water Nitrate and Adverse Reproductive Outcomes: A Systematic PRISMA Review' is a narrative meta-analysis (review) on the impact of nitrates/nitrites on reproductive outcomes. Methodology of a systematic literature search is well described. Technically, the manuscript is well written. However, there are not many original studies available on such particular topic, from this reason only one study in humans and 11 very heterogeneous animal studies were included. The weak point of the manuscript is that the only conclusion is that there are almost no studies and those few existing studies were scored to be of low quality.
Comments and Suggestions for Authors
* I suggest you extend the introduction with more detail description why nitrates/nitrites could have impact on reproductive outcomes.
* The sources of exposure to nitrates are very complex. It is questionable if drinking water is the major one.
* The concept of the paper is that nitrates must have adverse reproductive outcomes and the only study in humans shows decreased risk of spontaneous abortion after exposure to nitrates/nitrite. One would say the impact of nitrates intake is beneficial.
Reviewer 2 Report
- Abstract: “However” does not need to be capitalized (line 2).
- Introduction: How prevalent is nitrate contamination in drinking water worldwide? Is it high in certain geographic regions? Associations between nitrate and other adverse birth outcomes (birth weight, preterm birth)?
- Methods: For future reference, I recommend the Navigation Guide, specifically designed for environmental exposures and pregnancy outcomes, with input from human and animal advice (Woodruff TJ, Sutton P. The Navigation Guide systematic review methodology: a rigorous and transparent method for translating environmental health science into better health outcomes. Environ Health Perspect. 2014;122(10):1007-1014. doi:10.1289/ehp.1307175).
- Results: I would be more explicit in the text about how you present the results in the tables. For example, “Study characteristics, results, and the assigned score from quality assessment are presented… for the human studies in women (Table 1), animal studies in females (Table 2) and animal studies in males (Table 3).
- Results (p.13, line 170, 181, 1830: Paragraphs with one sentence should be combined where possible.
- Results (p. 14, line 208): When reporting results, reserve judgement for the discussion. For example, “A score of “high risk of bias” was only given once to two studies.” Remove “only”.
- The scope of the review could have been broadened to include additional adverse reproductive outcomes such as adverse birth outcomes (stillbirth, preterm birth, low birth weight), to include a better picture of the potential harmful reproductive outcomes of nitrate. This would have enabled the authors summarize the human literature on the topic.
